# MASKINVERSION: LOCALIZED EMBEDDINGS VIA OPTIMIZATION OF EXPLAINABILITY MAPS

## ABSTRACT

Vision-language foundation models such as CLIP have achieved tremendous results in global vision-language alignment, but still show some limitations in creating representations for specific image regions. To address this problem, we propose MaskInversion, a method that leverages the feature representations of pre-trained foundation models, such as CLIP, to generate a context-aware embedding for a query image region specified by a mask at test time. MaskInversion starts with initializing an embedding token and compares its explainability map, derived from the pretrained model, to the query mask. The embedding token is then subsequently refined to approximate the query region by minimizing the discrepancy between its explainability map and the query mask. During this process, only the embedding vector is updated, while the underlying foundation model is kept frozen allowing to use MaskInversion with any pre-trained model. As deriving the explainability map involves computing its gradient, which can be expensive, we propose a gradient decomposition strategy that simplifies this computation. The learned region representation can be used for a broad range of tasks, including open-vocabulary class retrieval, referring expression comprehension, as well as for localized captioning and image generation. We evaluate the proposed method on all those tasks on several datasets such as PascalVOC, MSCOCO, RefCOCO, and OpenImagesV7 and show its capabilities compared to other SOTA approaches.

## 1 INTRODUCTION

Foundation models such as CLIP (Radford et al., 2021), pre-trained with a contrastive loss on large-scale image-text datasets, have significantly advanced vision-language understanding. However, those models focus on a global vision-language alignment in training, matching the respective text and image class ([CLS]) tokens, thus only the globally pooled information. As a result, such models often struggle with tasks requiring precise localization or the recognition of specific image regions, necessitating novel approaches to harness their full potential. In the following, we tackle the problem of generating embeddings localized to specific image regions from pretrained vision-language models. While it is possible to obtain such embeddings via naïve solutions, e.g. by processing only the cropped region, or aggregating the local token embeddings over a mask, such simple approaches often do not yield optimal results: cropping can remove important context, while token aggregation over region features might not result in a good, aligned representation as local tokens do not always correspond to the correct representation (Zhou et al., 2022).

Different approaches have been proposed to address the problem of localized vision-language tasks: ReCLIP (Subramanian et al., 2022) uses colored boxes during training to localize the alignment between vision and language. Fine-grained visual Prompting (FGVP) employs different masking strategies to force the model to focus on the relevant object region. AlphaCLIP (Sun et al., 2024) finetunes CLIP together with an alpha channel to highlight the region of interest. Finally, RIS (Yu et al., 2023) proposes a token masking pipeline to achieve zero-shot referring image segmentation.

Following this line of works, we propose MaskInversion, inspired by Text Inversion (Gal et al., 2023), as a method to learn a localized embedding for a query image region specified by a mask at test time. MaskInversion differs from previous methods as it does not adapt the vision-language backbone, but instead leverages the explainabilty map of a frozen backbone at test-time to optimize a representation, namely a token that captures the localized embedding (localized embedding token), for a given region

Figure 1: **MaskInversion Applications:** The proposed MaskInversion method generates a localized embedding without modifying the vision encoder, thereby enabling seamless integration as a drop-in replacement for the vision encoder output across various scenarios. *(Localized Classification):* classify each region of an image independently. *(Localized Captioning):* direct the attention of an LLM to specific parts of an image. *(Localized Diffusion):* used in conjunction with a diffusion model, generates variations of specific regions of images. All applications demonstrated here can be achieved by simply replacing the original vision encoder with MaskInversion, without further tuning.

mask. We start with initializing the localized embedding token from the global class token produced by CLIP. This token representation is then used to compute the initial explainability map for its current representation. We then compute the difference between the explainability map and the query mask. The token representation is then subsequently updated so that its representation generates an explainability map that matches the query mask. In this manner, we learn a token representation specific to the image region covered by the query mask.

Note that the token representation learning process is done for each mask separately. Thus, several different localized embedding tokens are created from the same image when multiple object masks are given. We can further enhance the computational efficiency for this case by exploiting the fact that the derivation of the explainability map is fixed because of the frozen backbone, and is independent of a query mask. Namely, we propose a gradient decomposition strategy that simplifies the gradient computation associated with the explainability method. Finally, while the resulting region-based localized embedding tokens are optimized for their specific mask, it can sometimes be desirable to also include global context. While e.g. for classification it does not matter if a bicycle is leaned to a tree or floating in the sky, such context information can be critical for referring expressions. We therefore further propose an add-on regularization loss that aligns the learned representation to the global image representation and allow to balance between global and local representations if needed.

The resulting localized embeddings can be used in various downstream tasks as shown in Figure 1, including region-based localized classification, region-based localized captions (as in AlphaCLIP), and localized image generation. In all cases, we assume a zero-shot setting and use our localized embedding tokens as a drop-in replacement, e.g. for the CLIP ViT [CLS] token. This means e.g. for region-based zero-shot classification that we compute the localized embedding token and match it with the respective class prompts, e.g. "A photo of a dog". We evaluate the proposed method in all those scenarios, showing improved performance compared to other methods in each domain.

We summarize the contributions of our work as follows: (1) Given an image and a query mask, we learn a localized embedding at test time that captures the region characteristics within the mask in a single token. The learned can be used as a drop-in replacement for any application based on the same backbone. (2) We propose gradient decomposition to make the process computationally efficient for multiple query masks in the same image. (3) We evaluate the resulting representation on various region-based downstream tasks, showing improved results across a range of different applications.

## 2 RELATED WORK

**Localized Representation Learning** The task of enhancing the localized embedding of foundation models such as CLIP (Radford et al., 2021) has gained increased attention recently. While these models, trained on noisy image-text pairs scraped from the internet, have proven to be a rich source

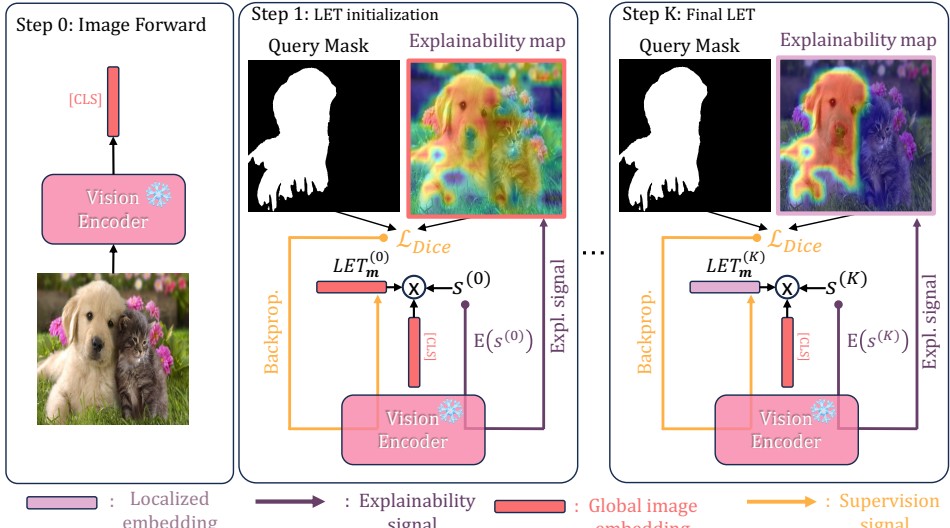

Figure 2: **MaskInversion:** *(Step 0)*: the input image is forwarded only once during the whole MaskInversion process. *(Step 1):* the localized embedding ($LET_{\mathbf{m}}$) is initialized to the vision encoder's [CLS] token. Then the $LET_{\mathbf{m}}$ embedding is trained such that its explainability map is similar to the query mask. *(Step K):* after $K$ gradient descent iterations, we obtain the final localized embedding $LET_{\mathbf{m}}$ that can be used for downstream task.

of supervision for learning a broad range of concepts (Radford et al., 2021), their training methodology, which matches the global feature representation of an entire image with its corresponding caption, often falls short in the context of localized tasks. ReCLIP (Subramanian et al., 2022) uses a combination of clipping and blurring to receive a region-specific embedding and further tries to capture relations between those instances. Shtedritski (Shtedritski et al., 2023) found that a red circle around an object can direct the model's attention to that region, thus producing a 'localized' CLS token while maintaining global information. As an extension to those works, Yang et al. (Yang et al., 2024) explore different techniques for Fine-Grained Visual Prompting (FGVP), including outlining the relevant object or blurring the rest of the image (Blur Reverse Mask) and using the resulting CLIP CLS token for various downstream tasks. We find that especially the masked blurring provides a strong baseline. Another line of work, CPT (Yao et al., 2024) fine-tunes an existing language model to allow for a prompting based on different color patches. AlphaCLIP (Sun et al., 2024) takes a similar approach by retraining CLIP to take an alpha mask alongside the original image as input, focusing the model's output feature representation on the area covered by the alpha mask. However, this method requires millions of mask annotations to generalize effectively. Note that MaskInversion differs from both streams of work: from current visual prompt tuning methods, as it does not seek to change the input image directly to get a localized CLS token embedding, but instead learns a new representation for the given maks, but also from methods that rely on masked-based pertaining as MaskInversion is applied at test time and does not assume any adaptation of weights of the frozen backbone. Finally, Gal et al. (Gal et al., 2023) proposed text inversion as an idea related to capture embeddings, but for the case of learning a token that represents a certain object to be injected into a text-to-image generator. While this idea is the conceptual inspiration for this work, MaskInversion differs from this method as it captures regional properties via binary masks and respective explanation maps, while text inversion focuses on learning general object properties from multiple images.

**Explainability Methods** The proposed MaskInversion method relies on the use of explainability methods to guide the model to focus on the desired area in the image. These methods explain model decisions by assigning a score to each image pixel representing its importance to the model's output. Gradient-based methods, which compute explanations based on the gradient of the model's prediction with respect to the model output, are computationally efficient and easy to understand since they are a direct function of the model's parameters and do not rely on additional models or image modifications. They have been used successfully to identify reasoning, spurious correlation, and trustworthiness in traditional computer vision models (Erhan et al., 2009; Simonyan et al., 2014; Springenberg et al., 2015; Sundararajan et al., 2017; Selvaraju et al., 2017; Smilkov et al., 2017; Kapishnikov et al.,

2019). Furthermore, gradient-based methods are differentiable, making it possible to use them as an objective function. For instance, (Chefer et al., 2022) uses the explainability map to supervise the model training, enforcing the model to base its classification prediction on the part of the image that contains the object, thus enhancing the model's robustness. Similarly, (Paiss et al., 2022) leverages the explainability signal to force an image generation model to utilize the entirety of the text prompt given by the user. While early explainability methods were developed for Convolutional Networks, with perhaps the most known one being GradCAM (Selvaraju et al., 2017), the widespread use of ViTs has led researchers to adapt existing methods or develop new ones specifically for transformers. For instance, rollout (Abnar & Zuidema, 2020) combines all the attention maps via matrix multiplication to trace the flow of importance through the transformer's layers. Chefer et al. (Chefer et al., 2021) extended rollout by weighting the attention by their gradient, making the method class-specific. Recently, LeGrad (Bousselham et al., 2024) proposed a gradient-based feature-attribution method specifically designed for ViT architectures. The method relies solely on the gradient of the attention maps, making it fast and easy to use. We chose LeGrad as the default explainability method used in the evaluation, but note that MaskInversion is a general method and can be used in conjunction with any differentiable explainability method.

## 3 METHOD

The proposed method, coined as *MaskInversion*, aims to learn a localized embedding or feature vector that encapsulates an object's characteristics within an image specified by a query mask. This embedding should not solely represent the object's intrinsic properties but also capture the broader context of the entire image. For instance, the embedding of a mask of a cat should differ when the cat is situated in an empty field or when it is crossing a bustling road. To achieve this, we utilize representations provided by foundation models, such as CLIP. Our approach learns a token that captures the foundation model's feature representation on the image region specified by the mask. Hence, the foundation model remains fixed during our process.

As shown in Figure 2, we start with the initialization of an embedding vector that serves as a localized embedding token of the mask. This vector is then refined through an iterative optimization process guided by an explainability map generated from the foundation model. The explainability map provides a visual indication of the areas within the image that are most influential on the initial embedding, thereby allowing for targeted refinement. The optimization process is supervised by enforcing the generated explainability map to be similar to the query mask. We can optionally use a regularization loss to ensure the mask embedding is congruent with the model's learned manifold. Finally, we improve the computational load for this process, especially for the case of computing multiple embeddings based of different masks for the same image, via gradient decomposition.

### 3.1 BACKGROUND/PRELIMINARIES: EXPLAINABILITY METHODS

The proposed MaskInversion method relies on the use of explainability methods to guide the creation of the localized embedding token. Here, we give a brief introduction to explainability methods, focusing on "gradient-based" methods (e.g. GradCAM(Selvaraju et al., 2017)). We let $\mathcal{F}$ denote a model that maps an input image $x \in \mathbb{R}^{3 \times W \times H}$ to an output an activation $\mathcal{F}(x) = s \in \mathbb{R}$ . In practice, $s$ could be derived from a classifier's score for a particular class or the cosine similarity between image and text embeddings in a vision-language model (e.g., CLIP). For a given layer $l \in \{1, \ldots, L\}$ of $\mathcal{F}$, we denote $\mathbf{A}^l$ the intermediate representation of the model. $\mathbf{A}^l$ can be intermediate features maps in the case of CNNs(Selvaraju et al., 2017), intermediate tokens or attention maps in the case of Vision Transformers(Dosovitskiy et al., 2021). We also denote the partial derivative of the activation $s$ w.r.t $A^l$ as $\nabla A^l = \frac{\partial s}{\partial A^l}$.

Gradient-based explainability methods can be generally formulated as combination of operations between the intermediate representation $\mathbf{A} = (A^1, \ldots, A^L)$ and the gradients $\nabla \mathbf{A} = (\nabla A^1, \ldots, \nabla A^L)$: and produces a 2D heatmap denoted, $E = g(\mathbf{A}, \nabla \mathbf{A}) \in \mathbb{R}^{W \times H}$. For instance, in GradCAM (Selvaraju et al., 2017), $E$ is defined as $E(\mathbf{A}, \nabla \mathbf{A}) = \text{ReLU}\left(\sum_k \alpha_k \cdot \mathbf{A}_k^L\right)$, where $\alpha_k = \sum_{ij} \nabla A_{k,i,j}^L$ are the weights for the feature maps $\mathbf{A}^L$.

In the context of Vision Transformers (ViTs), we employ LeGrad (Bousselham et al., 2024). It focuses on the attention mechanism's role in aggregating information into the [CLS] token, which is crucial for ViTs. It considers the intermediate representations $\mathbf{A}^l$ to be the attention maps of the

self-attention layers. For a given activation score $s$, the gradient $\nabla A^l$ of $s$ with respect to the attention map $A^l$ is computed, and a ReLU function is applied to discard negative contributions:

$$\hat{E}^l(s) = \frac{1}{hn} \sum_h \sum_i \text{ReLU} \left( \frac{\partial s}{\partial \mathbf{A}_{h,i,.}^l} \right).$$ (1)

where $h$ is the number of heads and $n$ is the number of visual tokens. Then the explainability maps of each layers are averaged: $\bar{E} = \frac{1}{L} \sum_l \hat{E}^l(s)$. The final explainability map is then obtained by isolating the influence of the patch tokens, reshaping it into a 2D map, and applying min-max normalization to scale the scores between 0 and 1: $E = \text{norm}(\text{reshape}(\bar{E}))$. In practice, we utilize only the last attention map of the last layer to reduce computational cost.

## 3.2 Localized Embedding Learning via Explainability Map Optimization

The inputs to our method are an image $x \in \mathbb{R}^{3 \times W \times H}$ and a binary query mask $\mathbf{m} = (m_{i,j}) \in \mathbb{R}^{W \times H}$, $m_{i,j} \in \{0, 1\}$, specifying a region of interest. Our objective is to derive a localized embedding token $LET_\mathbf{m} \in \mathbb{R}^d$ that generates an explainability map that corresponds to the masked region.

**Embedding Token Initialization .** We initialize the localized embedding token $LET_\mathbf{m}^{(0)}$ by copying the global [CLS] token produced by the foundation model, $LET_\mathbf{m}^{(0)} = z^0 \in \mathbb{R}^d$. We then compute the cosine similarity between the embedding token and the average of the [CLS] and all patch tokens following (Bousselham et al., 2024) as the activation score for the explainability map:

$$s^{(0)} = \mathbf{cos} \left( LET_\mathbf{m}^{(0)}, \bar{\mathbf{z}} \right) \in \mathbb{R},$$ (2)

where $\bar{\mathbf{z}} = \frac{1}{n} \sum_p z_p$ represents the patch and [CLS] token of the ViT averaged across the spatial dimensions, and $\mathbf{cos}$ denotes the cosine similarity. Following the process described in Section 3.1, the score is used to compute the explainability map denoted as $\mathbf{E}^{(0)} = E(s^{(0)}) \in \mathbb{R}^{W \times H}$, with each element $\mathbf{E}_{i,j}^{(0)} \in [0, 1]$. This map $\mathbf{E}^{(0)}$ indicates the regions within the image that the initial embedding $LET_\mathbf{m}^{(0)}$ predominantly focuses on. Since the localized embedding is initialized with the [CLS] token our initial explainability map corresponds to the explainability map of the [CLS] token.

**Embedding Token Optimization.** To refine the initial guess and guide the embedding token representation towards the query mask, we treat the mask localized embedding $LET_\mathbf{m} \in \mathbb{R}^d$, corresponding to the query mask $\mathbf{m}$, as a *learnable vector* with $d$ parameters. We supervise the learning of this vector by optimizing its parameters for $K$ steps, with $k \in \{0, .., K\}$ so that, for each optimization step $k$, the resulting explainability map $\mathbf{E}^{(k)}$ for this token resembles the query mask $\mathbf{m}$. We achieved this through iterative gradient descent. Specifically, we quantify the discrepancy between the explainability map and the query mask using a soft Dice loss, as commonly employed in segmentation tasks (Milletari et al., 2016; Cheng et al., 2021) measuring region similarity:

$$\mathcal{L}_{\text{Dice}} = 1 - \frac{2 \times \text{intersection}(\mathbf{E}^{(k)}, \mathbf{m})}{\text{union}(\mathbf{E}^{(k)}, \mathbf{m}) + \epsilon},$$ (3)

where $\text{intersection}(\mathbf{E}^{(k)}, \mathbf{m})$ and $\text{union}(\mathbf{E}^{(k)}, \mathbf{m})$ are the intersection, realized by elementwise multiplications, and union, realized by elementwise addition, of the explainability map and the binary mask, respectively, and $\epsilon$ is a small constant to avoid division by zero. The Dice loss is minimized by optimizing the localized embedding $LET_\mathbf{m}$ parameters over $K$ iterations of gradient descent to yield the final embedding $LET_\mathbf{m} = LET_\mathbf{m}^{(K)}$.

**Regularization Loss.** The method, as described so far, will capture the representation of the indicated region. This can lead to the effect that the final representation $LET_\mathbf{m}$ is less aligned with the image itself, thus discarding any image information. But it can sometimes be helpful to have both a good region representation together with general image context. We, therefore, propose an add-on auxiliary regularization loss that forces the localized token embedding $LET_\mathbf{m}^{(k)}$ at each step $k$ to remain within the manifold of the image encoder:

$$\mathcal{L}_{\text{reg}} = 1 - \mathbf{cos} \left( LET_\mathbf{m}^{(k)}, z_0^L \right).$$ (4)

The final loss function is a weighted sum of the Dice loss equation 3 and the regularization loss:

$$\mathcal{L} = \mathcal{L}_{\text{Dice}} + \alpha \cdot \mathcal{L}_{\text{reg}}, \tag{5}$$

where $\alpha \in \mathbb{R}$ is a hyperparameter that modulates the influence of the regularization loss. It allows us to regulate how much region vs. global information should be encoded in the output token embedding. We found that this sepcifically helps for tasks that need context knowledge such as referring expressions, while 'object-only' tasks such as region-based/localized classification do not profit from such an alignment, thus setting $\alpha = 0$ for those cases.

## 3.3 FASTER MASK INVERSION VIA GRADIENT DECOMPOSITION

The derivation of the explainability map necessitates the calculation of a gradient, and similarly, each gradient descent iteration requires the computation of a gradient with respect to the loss function $\mathcal{L}$. Consequently, this iterative process requires the evaluation of second-order derivatives of the form $\frac{\partial \mathcal{L}}{\partial LET_{\mathbf{m}}^{(k)}}(LET_{\mathbf{m}}^{(k)}, \nabla \mathbf{A})$, which can be computationally intensive and numerically unstable.

To enhance the computational efficiency of this process, it is advantageous to obviate the need for backpropagation to generate explainability maps at each iteration. We propose a gradient decomposition strategy that simplifies the gradient computation associated with the explainability method. For a given iteration $k$, the gradient decomposition can be expressed as follows:

$$\nabla \mathbf{A} = \frac{\partial s}{\partial \mathbf{A}} = \frac{\partial \bar{\mathbf{z}} \cdot \left(LET_{\mathbf{m}}^{(k)}\right)^T}{\partial \mathbf{A}} = \frac{\partial \bar{\mathbf{z}}}{\partial \mathbf{A}} \cdot \left(LET_{\mathbf{m}}^{(k)}\right)^T \in \mathbb{R}^{h \times n \times n} \tag{6}$$

where h is the number of heads and n is the number of visual tokens. This equation holds true because the mask $LET_{\mathbf{m}}^{(k)}$ is not dependent on the activations $\mathbf{A}^L$. By decomposing the gradient in this manner, the task of generating the explainability map transitions from a gradient computation to a dot product operation between $LET_{\mathbf{m}}^{(k)} \in \mathbb{R}^d$ and $\frac{\partial \bar{\mathbf{z}}}{\partial \mathbf{A}} \in \mathbb{R}^{h \times n \times n \times d}$. As a result, the proposed gradient decomposition approach significantly reduces the computational load by eliminating the need to compute the gradient of the score function $s$ with respect to the activations $\mathbf{A}$ multiple times. Instead, a single computation of the gradient $\frac{\partial \bar{\mathbf{z}}}{\partial \mathbf{A}}$ suffices for all subsequent gradient descent steps, thereby expediting the mask inversion process and enhancing its numerical stability.

## 4 EXPERIMENTS

### 4.1 DOWNSTREAM TASKS

In the following, we will give a brief overview of these tasks, their metrics and datasets. Please see the Appendix B for all details.

**Referring Expressions** To assess the proposed method's ability to capture localized properties, we evaluate it for referring expression classification. Given an image and a set of masks, we generate an embedding for each mask within an image and match the generated region embeddings to a set of text queries (referring expressions) encoded with the respective text encoder. The query mask whose localized embedding exhibits the highest cosine similarity with the text embedding is selected. We employ standard referring expression datasets: PhraseCut (Wu et al., 2020), RefCOCO, and RefCOCO+ (Kazemzadeh et al., 2014), reporting top-1, top-5, top-10 accuracy, mean Intersection over Union (mIoU) and overall Intersection over Union (oIoU).

**Class Retrieval** Zero-shot classification requires classifying an image by matching its visual embedding with the textual description of the classes present in the dataset. Here, we propose to increase the granularity by using it to *classify a specific region* of the image: given a query mask of an object, classify it by matching its localized embedding to the text embeddings of the classes in the datasets. For this, we leverage two semantic segmentation datasets, PascalVOC (Everingham et al., 2015) and PascalContext (Mottaghi et al., 2014), and one instance segmentation dataset, MSCOCO (Lin et al., 2014). The performance is evaluated using the top-1, top-5, and top-10 accuracy. Finally, we challenge the proposed method in a large-scale open-vocabulary setting. We utilize a subset of the OpenImagesV7 (Benenson & Ferrari, 2022), which offers mask annotations for a diverse array of objects across 350 unique classes.

| | Method | zero-shot | PhraseCut | | | RefCOCO | | | RefCOCO+ | | |
|---|---|---|---|---|---|---|---|---|---|---|---|
| | | | Acc@1 | Acc@5 | Acc@10 | Acc@1 | mIoU | oIoU | Acc@1 | mIoU | oIoU |
| CPT ‡ | RN50x16 + ViT-B/32 | ✓ | - | - | - | 32.2 | - | - | 31.9 | - | - |
| GradCAM‡ | RN50x16 + ViT-B/32 | ✓ | - | - | - | 42.9 | - | - | 47.8 | - | - |
| ReCLIP‡ | RN50x16 + ViT-B/32 | ✓ | - | - | - | 45.8 | - | - | 47.9 | - | - |
| RedCircle‡ | RN50x16 + ViT-L/14@336 | ✓ | - | - | - | 49.8 | - | - | 55.3 | - | - |
| FGVP‡ | RN50x16 + ViT-B/32 +ViT-L/14@336 | ✓ | - | - | - | 52.9 | - | - | 57.4 | - | - |
| AlphaCLIP ‡ | ViT-B/16+ViT-L/14 | ✗ | - | - | - | 55.7 | - | - | 55.6 | | |
| RIS | ViT-B/32 | ✓ | - | - | - | - | - | 42.6 | - | - | 37.1 |
| CLIP* | ViT-B/16 | ✓ | 14.4 | 66.4 | 87.1 | 18.3 | 18.9 | 15.3 | 18.4 | 19.0 | 15.4 |
| Crop* | ViT-B/16 | ✓ | 15.1 | 67.0 | 87.6 | 17.9 | 18.5 | 15.5 | 19.0 | 19.5 | 16.1 |
| Masked Crop* | ViT-B/16 | ✓ | 48.3 | 89.7 | 97.2 | 52.3 | 52.9 | 41.2 | 58.7 | 59.4 | 47.5 |
| RedCircle* | ViT-B/16 | ✓ | 21.5 | 72.3 | 90.3 | 42.5 | 43.2 | 32.7 | 42.5 | 43.3 | 33.5 |
| FGVP* | ViT-B/16 | ✓ | 35.9 | 83.5 | 95.2 | 42.6 | 43.2 | 33.3 | 48.0 | 48.7 | 38.0 |
| AlphaCLIP* | ViT-B/16 | ✗ | 34.0 | 80.0 | 93.6 | 43.4 | 44.0 | 38.1 | 44.2 | 44.7 | 39.7 |
| MaskInversion | ViT-B/32 | ✓ | 54.8 | 93.0 | 98.5 | 54.1 | 54.7 | 42.3 | 55.8 | 56.5 | 44.3 |
| MaskInversion | ViT-B/16 | ✓ | 57.2 | 93.3 | 98.3 | 56.1 | 56.8 | 44.5 | 58.3 | 59.0 | 46.5 |
| MaskInversion | ViT-L/14 | ✓ | 60.2 | 94.9 | 98.7 | 56.1 | 56.7 | 42.0 | 60.2 | 60.9 | 47.5 |
| MaskInversion | ViT-H/14 | ✓ | **64.0** | **96.0** | **99.2** | **61.2** | **61.8** | **47.5** | **65.0** | **65.7** | **52.6** |

Table 1: Comparison with baselines on Referring Expression Retrieval. Given a query mask, the task is to retrieve the corresponding expression. ‡ indicates deviating evaluation settings where a pretrained region proposal is used, in that setting if the matched region has an IoU > 0.5, the prediction is counted as a hit; note that in this setting, several proposals could result in a hit. ∗ indicates reproduced results.

**Localized Captioning** Traditionally, image captioning models generate captions for entire images based on the visual representation provided by an image encoder. In contrast, we aim to evaluate our method's ability to focus the captioner on a specific image region while maintaining contextual relevance. To this end, we leverage a pretrained image captioner, CLIPCap (Mokady et al., 2021), and provide it with the localized embedding token of a query mask to generate a caption. CLIPCap is trained on top of the CLIP vision encoder and feeds its [CLS] token to GPT-2(Radford et al., 2019) to produce a caption. Here, we feed the localized embeddings of MaskInversion as a drop-in replacement of the CLIP [CLS] token to the captioner *without any finetuning*. As no dataset directly supports this evaluation type, we adapted an existing dataset, PhraseCut. To quantitatively evaluate the generated localized captions, we match the generated caption to the set of ground truth referring expressions for this image using the text encoder from CLIP (ViT-L/14 by OpenAI). We consider the caption correct if the cosine similarity between the generated caption and the ground truth referring expression for this mask is the highest. The reported metric for this task is the top-1 accuracy.

**Implementation Details** The proposed method is evaluated using pretrained CLIP vision-language models. For ViT-B/32, ViT-B/16, and ViT-L/14, we used the original weight from OpenAI (Radford et al., 2021), and for ViT-H/14, we used the weights `laion2b_s32b_b79k` from the OpenCLIP library (Cherti et al., 2023; Schuhmann et al., 2022). For the MaskInversion process, we use AdamW optimizer(Kingma, 2014) with 10 gradient descent iterations. For the loss equation 5, we set $\alpha$ to 5 for RefCOCO and RefCOCO+, and to 0 for all other datasets.

### 4.2 COMPARISON TO STATE-OF-THE-ART

**Referring Expression Retrieval** Table 1 presents the results on referring expression datasets. For related approaches, as there is no directly comparable setting, we provide both, reported as well as reproduced results. Note that the original evaluation settings can vary for different methods. For reproduced results, indicated by *, we adapt the evaluation setting to the case where ground truth masks are used as described in Sec. 4.1. We used the code provided by the authors of each method, forward each image together with the groundtruth masks of MSCOCO, and match the resulting representation to the text embedding of the respective backbone. We further compare with the following baselines: *CLIP* refers to the general CLIP baseline by using the image CLS token, *Crop* uses the CLS token of cropped region by forwarding only this region through CLIP, and *Masked Crop* refers to forwarding the full image, but keeping only the masked region and replacing all other pixels with the average pixel value of the dataset. For PhraseCut, MaskInversion outperforms all entertained baselines, regardless of the model size. On RefCOCO and RefCOCO+, MaskInversion also achieves SOTA performance. In addition, MaskInversion performance scales well when the backbone size increases, establishing a new SOTA on every data set when ViT-H/14 is used.

**Class Retrieval** Table 2 compares MaskInversion to other methods for the case of zero-shot class retrieval, keeping the same setting as detailed under *Referring Expression Retrieval*. MaskInversion again performs well compared to other methods on semantic segmentation datasets, such as PascalVOC and PascalContext.Furthermore, MaskInversion also exhibits good performance on the

| | Method | PascalVOC | | | PascalContext | | | COCO | | | OpenImagesV7 | | |
|---|---|---|---|---|---|---|---|---|---|---|---|---|---|
| | | Acc@1 | Acc@5 | Acc@10 | Acc@1 | Acc@5 | Acc@10 | Acc@1 | Acc@5 | Acc@10 | Acc@1 | Acc@5 | Acc@10 |
| ViT-B/16 | CLIP* | 40.1 | 87.2 | 95.6 | 17.8 | 38.7 | 52.7 | 25.0 | 54.9 | 72.6 | 28.9 | 63.4 | 72.7 |
| | Crop* | 27.9 | 51.2 | 72.4 | 5.6 | 13.2 | 20.4 | 23.9 | 34.5 | 41.5 | 0.8 | 3.8 | 7.05 |
| | Masked Crop* | 75.0 | 91.4 | 96.4 | 40.4 | 65.9 | 75.8 | 38.2 | 57.7 | 65.2 | 33.8 | 61.9 | 73.7 |
| | RedCircle* | 47.5 | 92.9 | 97.7 | 21.3 | 45.0 | 57.4 | 28.8 | 63.0 | 77.3 | 40.5 | 75.8 | 84.5 |
| | AlphaCLIP* | 52.6 | 85.9 | 93.8 | 27.7 | 60.9 | 75.1 | 30.9 | 55.9 | 70.3 | 43.0 | 77.4 | 84.3 |
| | FGVP* | 71.8 | 93.6 | 98.3 | 32.6 | 58.9 | 72.4 | 35.9 | 62.2 | 72.6 | 39.4 | 75.6 | 84.6 |
| | RIS* | 78.0 | 95.2 | 98.1 | 38.1 | 62.7 | 74.3 | 43.6 | 65.3 | 72.4 | 34.5 | 66.5 | 75.8 |
| B/32 | MaskInversion | 79.5 | 96.4 | 98.8 | 46.7 | 74.9 | 84.6 | 38.0 | 65.8 | 78.4 | 42.6 | 78.8 | 86.6 |
| B/16 | MaskInversion | 85.4 | 96.4 | 98.8 | 58.1 | 83.7 | 90.5 | 44.7 | 71.6 | 83.0 | 46.3 | 80.4 | 87.9 |
| L/14 | MaskInversion | 91.0 | 99.1 | 99.8 | 59.0 | 86.3 | 92.5 | 56.0 | 84.2 | 91.4 | 48.7 | 81.0 | 88.1 |
| H/14 | MaskInversion | **93.5** | **99.4** | **99.7** | **61.8** | **86.0** | **91.8** | **63.7** | **88.3** | **93.5** | **51.2** | **85.2** | **91.4** |

Table 2: Comparison with baselines on Class Retrieval for Segmentation Datasets. Given a mask, the task is to retrieve the corresponding class. ∗ indicates reproduced results.

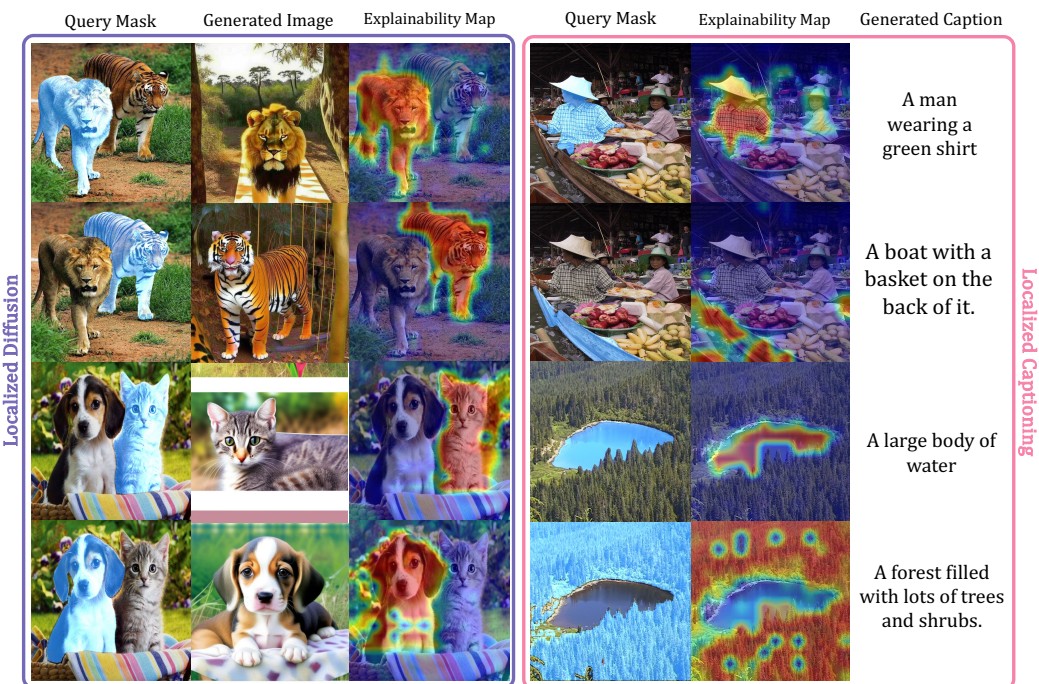

Figure 3: **Localized Embedding Visualizations:** Visualisation of the learned localized embedding using *(left)* a pretrained diffusion model; *(right)* an image captioner. In both cases, all the models are kept frozen and only the global feature representation of the vision encoder is replaced by the output of MaskInversion depending on the query mask.

instance segmentation dataset COCO. These results demonstrate that MaskInversion can effectively direct the attention of the foundation model to *multiple instances of the same object class at the same time, as well as to a single instance*. Here, MaskInversion also outperforms the recently proposed AlphaCLIP (Sun et al., 2024), which fine-tunes CLIP with millions of mask-text pairs annotations, thereby demonstrating its ability to excel without the need to fine-tune CLIP. Finally, looking at the results on OpenImagesV7, which features a significantly larger vocabulary of 350 classes, we can see that methods like AlphaCLIP, which are specifically trained for such tasks, perform well. However, MaskInversion still outperforms all other methods we compared, demonstrating its capability to handle large vocabularies.

## 4.3 LOCALIZED CAPTIONING ANALYSIS

**Localized captions** We further consider the performance of MaskInversion against CLIP and AlphaCLIP for localized captioning in Table 3. We use CLIPCap (Mokady et al., 2021) as the captioner and replace the CLIP image encoder with either AlphaCLIP or the output of MaskInversion without any fine-tuning. We observe that MaskInversion demonstrates the ability to focus the captioner on the area of interest, as the accuracy is

| Method | Acc |
|---|---|
| CLIP | 20.1 |
| AlphaCLIP | 31.8 |
| MaskInversion | **48.4** |

Table 3: **Localized captioning:** Given a query mask, the goal is to generate a caption that corresponds to the region highlighted by the mask. CLIPCap is used to generate the caption with CLIP-ViT-B/16.

more than doubled when using MaskInversion versus only using CLIP. Moreover, MaskInversion also significantly AlphaCLIP despite not involving any fine-tuning of the CLIP model.

**Qualitative Results** Figure 3 presents further qualitative examples of the localized captions generated by MaskInversion+CLIPCap for different query masks. These visualizations complement the quantitative benchmarks. The proposed method demonstrates a high degree of precision in focusing the captioning on specific image regions dictated by the query masks, as e.g. water and first are in are highly separated.

## 4.4 MASK EMBEDDING FOR IMAGE DIFFUSION

To further visualize the concepts captured in the learned representation output by MaskInversion, we employed $\lambda$-ECLIPSE (Patel et al., 2024), a state-of-the-art diffusion model. This model accepts a visual embedding from a ViT-bigG/14 CLIP model along with a text prompt, producing variations of the input image that correspond to the prompt. Utilizing the default settings of $\lambda$-ECLIPSE as described in (Patel et al., 2024), we conducted several experiments to generate images based on different query masks used for the MaskInversion process.

Figure 3 illustrates how the generated images vary depending on the mask used. This variation shows the effectiveness of MaskInversion in producing localized and contextualized embeddings. The images clearly focus on the objects or groups of objects within the bounds of the query mask, confirming that MaskInversion directs the model's attention to specific parts of the image. Moreover, we observe that the final explainability map generated by LeGrad(Bousselham et al., 2024) is focused on the area covered by the query mask, validating the effectiveness of the proposed optimization process.

| Mask Type | Acc |
|---|---|
| Mask | 44.7 |
| Erosion | 42.7 |
| Dilation | 44.3 |
| Box | 42.9 |
| Box + SAM | **45.0** |

Table 4: **Mask Quality Ablation:** assessment of the mask quality impact on MSCOCO for the Class Retrieval task.

## 4.5 ABLATIONS

**Impact of Mask Quality** MaskInversion utilizes an input query mask to direct the output of the foundation model toward the area covered by the mask. Given that the mask is a critical component of our method, it is imperative to assess how variations in mask quality affect MaskInversion's performance. To this end, we evaluate different mask conditions for the task of Class Retrieval on the MSCOCO dataset as shown in Table 4: *Box* uses the bounding boxes instead of precise segmentation masks, *Box+SAM* uses the bounding boxes to receive a mask via segmentation using SAM (Kirillov et al., 2023), and *Erosion* and *Dilation* apply the respective morphological operations to the original masks. Figure 7 shows qualitative examples for the different cases. Our findings indicate that eroding the mask leads to a more substantial decrease in performance compared to dilation. We further see a decrease in accuracy from $44.7\%$ to $42.9\%$ when using bounding boxes only, whereas the com-

| #Mask | Decomp. | Sec.↓ |
|---|---|---|
| 5 | ✗ | **0.10** |
| 5 | ✓ | 0.13 |
| 10 | ✗ | 0.15 |
| 10 | ✓ | **0.14** |
| 50 | ✗ | 0.65 |
| 50 | ✓ | **0.27** |
| 100 | ✗ | 1.27 |
| 100 | ✓ | **0.44** |

Table 5: **Gradient Decomposition Ablation:** Runtime using or not using gradient decomposition as described Sec.3.3 (ViT-B/16) for different number of masks, for 10 gradient descent steps.

bination of bounding boxes and SAM to derive the mask achieves comparable performance to the ground truth mask. This scenario is especially relevant for practical applications where users may find it easier to draw bounding boxes rather than detailed masks.

**Runtime Evaluation for Gradient Decomposition** Table 5 presents a runtime comparison of the vanilla MaskInversion, where the gradient gradient-based explainability map is computed at each iteration and for each mask, versus the "gradient-decomposition" proposed in section 3.3 for $K = 10$ steps. We observe that for any number of masks higher than 5 the proposed gradient decomposition is faster than the vanilla way of computing the explainability map (see appendix Sec. F for an ablation on the number of iterations).

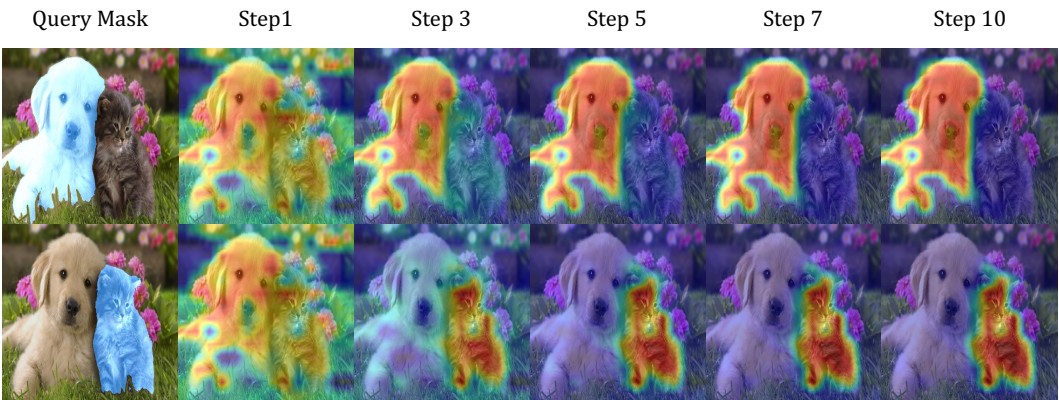

| Query Mask | Step1 | Step 3 | Step 5 | Step 7 | Step 10 |

Figure 4: **Visualization of the Explainability Maps throughout the optimization steps.**

### 4.6 OPTIMIZATION STEPS VISUALIZATION

Finally, Figure 4 provides a visualization of the explainability map throughout the optimization process employed by MaskInversion. It is observed that the explainability map increasingly concentrates on the region covered by the query mask as the optimization progresses. This observation is indicative of the method's ability to effectively focus the attention of the underlying foundation model on the designated areas of the image.

## 5 CONCLUSION

In this work, we proposed MaskInversion as a method to create region embeddings that are grounded in the rich feature representations of foundation models without the need to fine-tune the model. To this end, we leverage the concept of explainability maps to learn an embedding vector that is focused on a respective region. We extend this idea by an add-on regularization loss to balance global and local representations as well as by a gradient decomposition to improve runtime in case of multiple masks per image. This approach holds promise for many applications in computer vision, where understanding and manipulating specific regions of an image in relation to their context is important.

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

## A  APPENDIX

In the Appendix, we first provide additional details on the different downstream task in Sec.B. Sec.E provides visualizations of the mask distortion used for our ablations. SecF provides a more thorough ablation of the proposed gradient decomposition technique. Sec. G presents an ablation on the explainability method used for MaskInversion. Sec. H and I respectively discuss the limitations of SOTA methods as well as the proposed MaskInversion. Eventually, we provide additional qualitative examples of localized captioning and diffusion in Sec. J and Sec. K.

## B  DOWNSTREAM TASKS

**Referring Expressions**  To assess the proposed method's ability to capture localized properties, we evaluate it for referring expression classification. Given an image and a set of masks, we generate an embedding for each mask within an image and match the generated region embeddings to a set of text queries (referring expressions) encoded with the respective text encoder. The query mask whose localized embedding exhibits the highest cosine similarity with the text embedding is selected. We employ standard referring expression datasets, i.e. PhraseCut (Wu et al., 2020), RefCOCO, and RefCOCO+ (Kazemzadeh et al., 2014). For RefCOCO and RefCOCO+, we use the mask annotations from the MSCOCO (Lin et al., 2014) dataset, which has about 30 masks per image, thereby increasing the difficulty of the task. For PhraseCut, we consider the masks of all annotated referring expressions as candidates, reporting top-1, top-5, and top-10 accuracy. Additionally, following (Subramanian et al., 2022; Sun et al., 2024; Yang et al., 2024; Shtedritski et al., 2023), for RefCOCO and RefCOCO+, we report the mean Intersection over Union (mIoU) and overall Intersection over Union (oIoU).

**Class Retrieval**  Second, we consider the task of zero-shot classification as a common benchmark for vision-language models. In that task, an image is classified by matching its visual embedding with the textual description of the classes present in the dataset. Here, we propose to increase the granularity by using it to *classify a specific region* of the image: given a query mask of an object, classify it by matching its localized embedding to the text embeddings of the classes in the datasets. For this, we leverage two semantic segmentation datasets, PascalVOC (Everingham et al., 2015) and PascalContext (Mottaghi et al., 2014), with 19 and 59 classes, respectively, and one instance segmentation dataset, MSCOCO (Lin et al., 2014), with 80 classes. The performance is evaluated using the top-1, top-5, and top-10 accuracy metrics, denoted by $Acc@1$, $Acc@5$, and $Acc@10$. Finally, we challenge the proposed method in a large-scale open-vocabulary setting by using a dataset encompassing a substantially larger number of classes. We utilize a subset of the OpenImagesV7 (Benenson & Ferrari, 2022) dataset, which offers mask annotations for a diverse array of objects across 350 unique classes. The evaluation metrics are again top-1, top-5, and top-10 accuracy reported as $Acc@1$, $Acc@5$, and $Acc@10$.

**Localized Captioning**  Traditionally, image captioning models generate captions for entire images based on the visual representation provided by an image encoder. In contrast, we aim to evaluate our method's ability to focus the captioner on a specific image region while maintaining contextual relevance. To this end, we leverage a pretrained image captioner, CLIPCap (Mokady et al., 2021), and provide it with the localized embedding token of a query mask to generate a caption. CLIPCap is trained on top of the CLIP vision encoder and feeds its [CLS] token to GPT-2(Radford et al., 2019) to produce a caption. Here, we feed the localized embeddings of MaskInversion as a drop-in replacement of the CLIP [CLS] token to the captioner *without any finetuning*. As no dataset directly supports this evaluation type, we adapted an existing dataset, PhraseCut. To quantitatively evaluate the generated localized captions, we match the generated caption to the set of ground truth referring expressions for this image using the text encoder from CLIP (ViT-L/14 by OpenAI), consider the caption correct if the cosine similarity between the generated caption and the ground truth referring expression for this mask is the highest. The reported metric for this task is the top-1 accuracy.

## C  INFLUENCE OF $\alpha$

We conduct an extensive analysis of the hyperparameter $\alpha$ to understand its role in balancing local and global information within the learned embeddings. Figure 5 illustrates this effect through generated captions for different $\alpha$ values. When $\alpha = 0$, the model generates descriptions focused strictly on

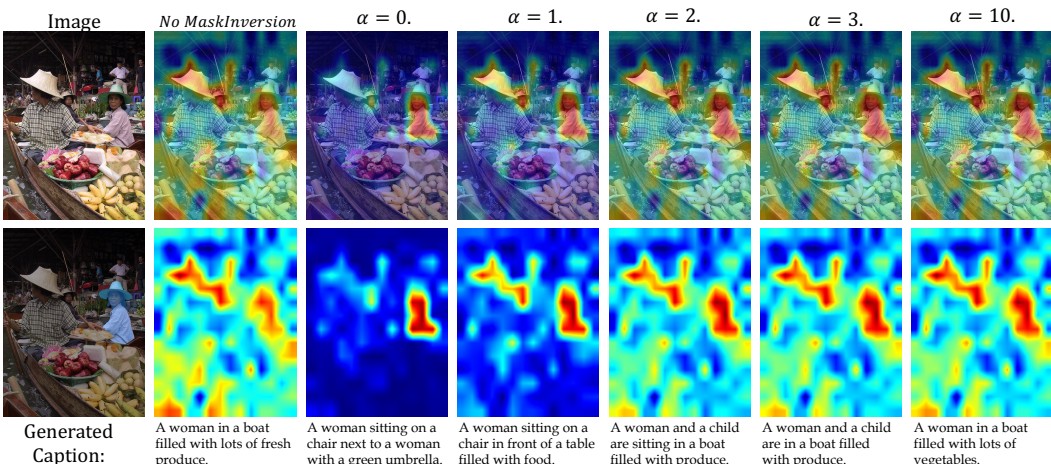

Figure 5: Qualitative analysis of the influence of $\alpha$ on the generated captions.

| alpha | 0.0 | 0.5 | 1.0 | 1.5 | 2.0 | 2.5 | 3.0 | 4.0 | 5.0 | 5.5 | 6.0 | 6.5 | 7.0 | 7.5 | 8.0 | 10.0 | 20.0 |
|-------|-----|-----|-----|-----|-----|-----|-----|-----|-----|-----|-----|-----|-----|-----|-----|------|------|
| Acc | 41.7 | 47.6 | 50.3 | 52.2 | 53.9 | 54.6 | 55.2 | 56.0 | 56.2 | 56.0 | 55.8 | 56.0 | 56.2 | 56.1 | 55.8 | 53.7 | 20.5 |

Table 6: Accuracy for different values of $\alpha$ on RefCOCO.

the masked region (*e.g.*, "woman in a boat"), while increasing $\alpha$ progressively incorporates more contextual information(*e.g.*, "produce" or "vegetables"). Quantitatively, we observe that performance on RefCOCO improves as $\alpha$ increases from 0 (41.7%) to an optimal value around $\alpha = 5.0$ (56.2%), before gradually declining for larger values. This sweet spot ($\alpha \approx 5.0$) represents an optimal balance where the embedding retains sufficient local information while leveraging beneficial contextual cues. Beyond $\alpha > 7.5$, performance deteriorates as the representation becomes increasingly similar to the global [CLS] token, with a dramatic drop at $\alpha = 20.0$ (20.5%). This analysis demonstrates that $\alpha$ effectively functions as a control mechanism for trading off local detail against global context in the learned representations.

## D  MULTI-OBJECT

While quantitative evaluation of multi-object scenarios presents inherent challenges, we demonstrate MaskInversion's capability to handle multiple objects through qualitative analysis. As shown in Figure D, our method effectively captures the relationships and context of multiple objects within a single mask. For instance, when given a mask covering multiple Pokémon characters, the generated diffusion outputs maintain coherent representations of all objects while preserving their spatial relationships and individual characteristics. The diffusion model successfully reconstructs multiple objects from the localized embedding, indicating that MaskInversion effectively encodes information about multiple entities and their relative positioning. This is particularly evident in cases where the mask encompasses groups of similar objects (e.g., multiple Pokémon) or diverse object combinations, demonstrating the method's robustness in handling complex, multi-object scenarios without losing individual object details or their contextual relationships.

## E  MASK QUALITY

Figure 7 provides a visualization of the different mask degradation settings entertained in Table 4.

## F  GRADIENT DECOMPOSITION

Figure 8 provides a more thorough comparison of the vanilla MaskInversion process described in Section 3.2 against the gradient decomposition trick described in Section 3.3. Namely, Figure 8 extends Table 5 to different numbers of gradient descent iterations and to more number of masks.

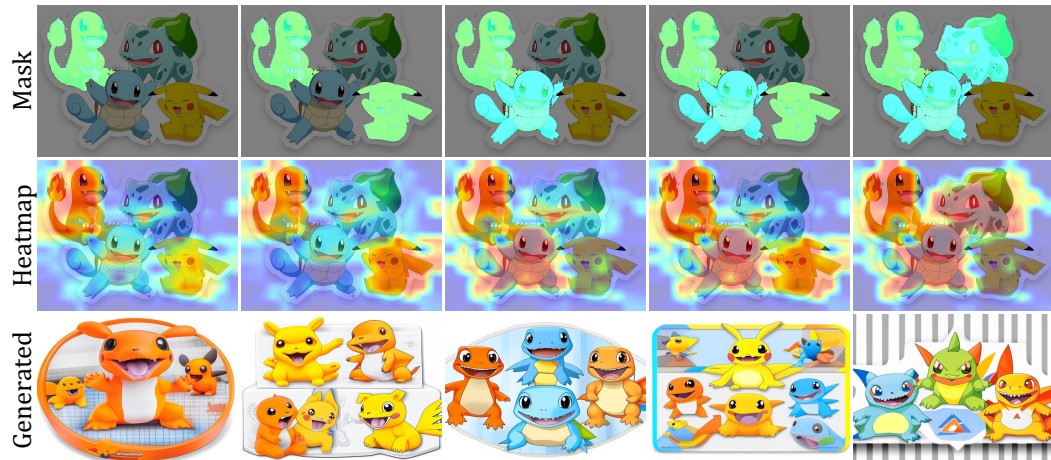

Figure 6: **Multi-Object Analysis:** Visualization of MaskInversion's ability to handle multiple objects. *(top)* Query masks highlighting different combinations of objects, *(middle)* corresponding heatmaps showing the model's focus regions, and *(bottom)* generated images using $\lambda$-ECLIPSE demonstrating the preservation of multiple object characteristics in the learned embeddings.

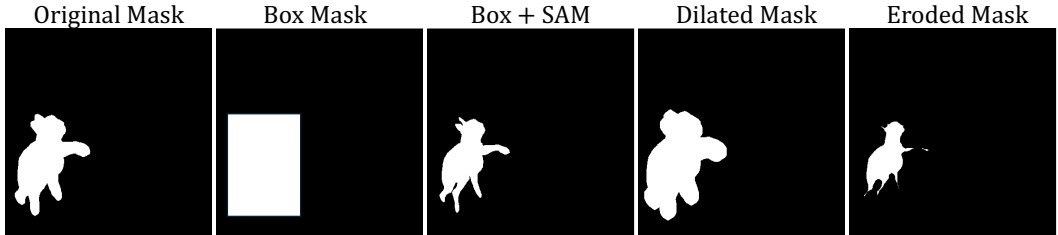

Figure 7: **Mask Quality Ablation:** example of different mask degradation settings.

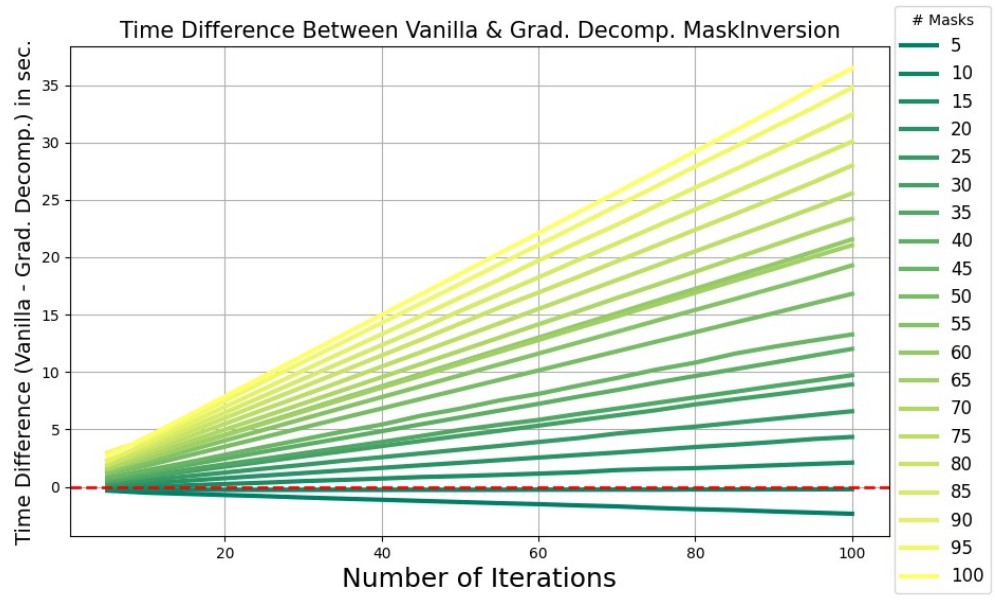

Figure 8: **Gradient Decomposition:** Time difference between using or not using the gradient decomposition technique described Sec.3.3, using ViT-B/16 for different numbers of masks and iterations ranging from 5 to 100. The time difference is in seconds.

## G  Impact of the Explainability Method

Given that MaskInversion leverages an explainability method
to guide the inversion process, its dependency on the choice of
explainability method was evaluated. We experimented with
alternative gradient-based methods, such as GradCAM and
CheferCAM, in place of the originally used LeGrad. The com-
parative results on the MSCOCO dataset are presented in Ta-
ble 7. LeGrad significantly outperformed the other methods,
which can be attributed to its design specificity for ViT archi-
tectures, unlike GradCAM and CheferCAM, which are tailored
for CNNs and general transformers, respectively. This find-
ing aligns with the observations in (Bousselham et al., 2024),
where LeGrad demonstrated superior localization capabilities essential for the tasks addressed by
MaskInversion. Thus, the selection of an appropriate explainability method is crucial for optimizing
the performance of MaskInversion.

| Expl. Method | Acc@1 |
|---|---|
| GradCAM | 34.6 |
| GradCAM‡ | 47.6 |
| CheferCAM | 12.6 |
| LeGrad | 85.4 |

Table 7: **Explainability Method Ablation:** MaskInversion perfor-
mance using different explainabil-
ity methods on the class retrieval
task on PascalVOC. ‡indicates a
modified version of GradCAM
without the ReLU operation.

## H  SOTA Methods' Limitations

Table 8 provides a description of the different baselines we compare MaskInversion to.

| Method | Finetune Model | Modify Img. | Description |
|---|---|---|---|
| Crop | ✗ | ✓ | Crop the input image, thus losing the context |
| RedCircle | ✗ | ✓ | Draw a red circle around the area of interest. Contingent on the biases in the training data and modifying the image can cause a domain gap. |
| Masked Crop | ✗ | ✓ | Crop the input image and mask the background. |
| FGVP(Yang et al., 2024) | ✗ | ✓ | Heavily blur the background, thus losing the context. |
| RIS(Yu et al., 2023) | ✗ | ✓ | Masks the features of the ViT after a certain number of layers to prevent the [CLS] token to aggregate information from outside the mask. |
| AlphaCLIP(Sun et al., 2024) | ✓ | ✗ | Finetunes CLIP to take as input an image and a mask. AlphaCLIP was trained on fine-grained mask/text pairs. |

Table 8: On one hand, directly modifying the input pixels can cause a domain gap between what the
model was trained on and what it is used for (e.g., RedCircle & Masked Crop). Moreover, it can also
completely remove the context that can be crucial for downstream tasks (e.g., Crop & Masking). On
the other hand, finetuning the model can not only result in forgetting the knowledge accumulated
during pretraining but also requires fine-grained mask/text data (*e.g.* AlphaCLIP). Also, the training
needs to be done for every model.

## I  Limitations

Firstly, the efficacy of MaskInversion is inherently tied to the availability and quality of explainability
methods that integrate well with the foundation model used. Models lacking robust explainability
frameworks may not fully benefit from the MaskInversion approach, as the method relies on accurate
and interpretable explanations to guide the inversion process. Consequently, the performance of
MaskInversion may degrade when applied to models with suboptimal explainability methods.

Secondly, foundational models like CLIP are often trained on using small-resolution images, usually
$224 \times 224$. This characteristic imposes a downstream limitation on the MaskInversion method,
particularly when the task involves focusing the model's attention on small objects within the image.
The reduced resolution can hinder the method's ability to accurately capture fine-grained details,
thereby affecting the overall performance in scenarios requiring high precision on small-scale features.
To mitigate that problem, in this work, we used bicubic interpolation on the pretrained positional
embedding of the ViT to increase the resolution at inference from $224 \times 224$ to $448 \times 448$.

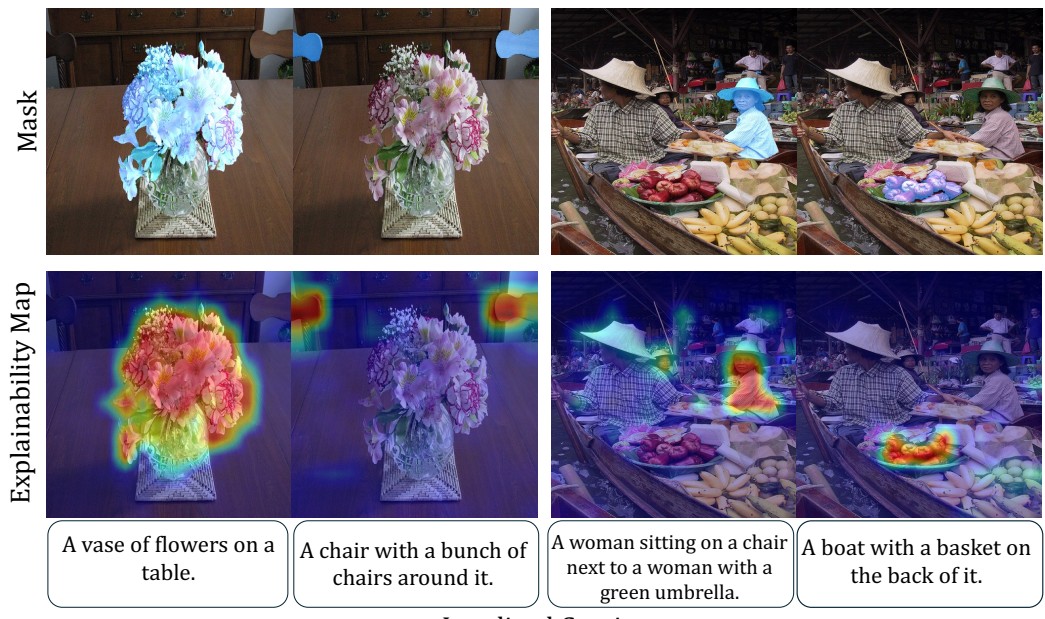

Figure 9: **Additional Localized Captions.**

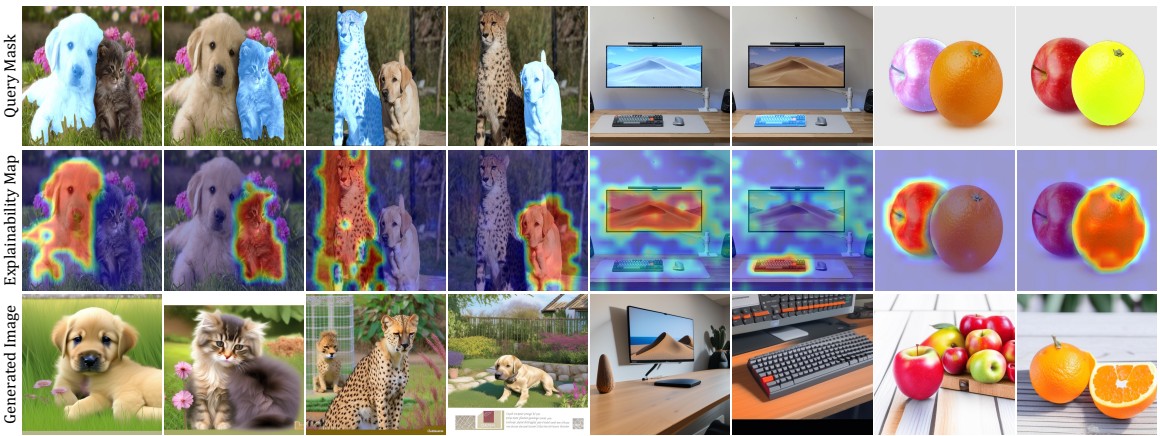

Figure 10: **Additional Localized Diffusion Examples.**

## J    ADDITIONAL LOCALIZED CAPTIONS

Figure showcases additional examples of localized captions for different masks as well as the final explainability map of the associated localized embedding. We observe that the generated caption essentially focuses on the area covered by the query mask, validating that the proposed MaskInversion is able to steer the visual focus toward the desired region.

## K    ADDITIONAL LOCALIZED DIFFUSION

Figure 10 provides additional visualization of the learned localized embedding for different mask queries. The visualization of the final explainability map is also provided. We observe that for each example the MaskInversion process is effectively able to steer the visual focus of the vision encoder toward the area of interest. Interestingly, when prompted with the mask of the monitor, the generated image contains a monitor with the same wallpaper scene, hence showcasing that the learned localized embedding learned a rich representation of the queried area.

