# OpenReview forum: "MaskInversion: Localized Embeddings via Optimization of Explainability Maps"
_ICLR.cc/2025/Conference — Submitted to ICLR 2025_

### Official Review · Reviewer_rbTp · 2024-11-02

**Soundness:** 3
**Presentation:** 3
**Contribution:** 3
**Rating:** 6
**Confidence:** 3

**Summary:**

The paper proposes a new method that uses explainability maps from pretrained models to generate localized embeddings. These embeddings can represent object properties while capturing the broader image context. The paper further demonstrates that these learned region representations are versatile and can be applied to various tasks, including retrieval, grounding, captioning, and image generation.

**Strengths:**

The paper introduces a novel approach leveraging explainability methods to enable the model to focus on specific regions within an image. Unlike traditional techniques like clipping, blurring, or masking, this approach allows the model to retain access to global image information. The method is clearly outlined and validated through comprehensive downstream tasks, demonstrating its effectiveness.

**Weaknesses:**

The paper primarily focuses on single-object scenarios, lacking analysis on multiple objects and their interactions. Including experiments and analysis on multi-object scenarios would strengthen the study and provide a more comprehensive evaluation of the method's effectiveness. For instance, datasets like MSCOCO, with complex captions involving multiple objects, could offer valuable insights; sharing examples from such datasets would further illustrate the model's performance in these scenarios.

**Questions:**

1. What type of global image context does this method capture? Could the authors provide visualizations, like attention map, to illustrate how the global context influences localized embeddings across different scenarios? This would clarify the method’s effectiveness in capturing and utilizing global context for downstream tasks.
2. In referring expression retrieval tasks, MaskInversion with ViT-B/16 underperforms compared to Masked Crop in RefCOCO+. Could the authors provide a detailed analysis investigating the reasons for this discrepancy?
3. Minor comment: In the related work section, "maks" should be corrected to "masks".

---

> ### Author Response · Authors · 2024-11-22
>
> Thank you for your thoughtful review and for highlighting the strengths of our approach, particularly regarding our novel use of explainability methods to enable region-specific focus while maintaining global context information. We appreciate your constructive feedback and questions, which we address below:
>
> # 1. Multi-object scenarios:
> We appreciate the reviewer's feedback regarding multi-object scenarios. While our presentation may have understated this capability, MaskInversion naturally handles multiple objects, as demonstrated quantitatively in our experiments on semantic segmentation datasets like PascalContext (Table 2), where masks frequently encompass multiple instances of the same object class.
>
> To further illustrate this capability qualitatively, we have added visualizations using λ-ECLIPSE diffusion model in **Figure 6**. of the Annex of the current paper version (please see the Annex written in blue).
> The figure shows how MaskInversion effectively captures multiple objects within a single mask, preserving their individual characteristics. For instance, when the mask covers multiple characters (as shown in the rightmost examples), the generated images accurately reflect the group composition while maintaining contextual relationships. This demonstrates that our method can effectively encode complex, multi-object arrangements without requiring explicit object-level separation. The explainability maps (top row) further validate this, showing how attention is appropriately distributed across multiple objects within the masked region. While quantitative evaluation of multi-object scenarios remains challenging due to the lack of standardized metrics, these qualitative results strongly suggest that MaskInversion successfully handles complex, multi-object compositions.
>
> # 2. Global context capture and visualization:
> To address your question about global context capture, we conducted additional experiments analyzing how the regularization parameter $\alpha$ influences the balance between local and global information (please see **Table 6** written in blue in the updated paper). We observe that:
> - Lower $\alpha$ values in [0,1] result in highly focused embeddings that primarily capture the masked region
> - Medium $\alpha$ values in [2,5] incorporate relevant contextual information while maintaining region specificity
> - Higher $\alpha$ values (>10) progressively approach the behavior of the global [CLS] token
> We have added visualizations in the paper (**Figure 5**) showing explainability maps and corresponding captions across different $\alpha$ values, demonstrating how this parameter effectively acts as a "slider" for controlling the local-global information balance. This is particularly valuable for tasks like referring expressions where contextual understanding is crucial.
>
> # 3. Performance on RefCOCO+:
> Regarding the performance difference between MaskInversion and Masked Crop on RefCOCO+ with ViT-B/16, this is an interesting observation that reveals important characteristics of the dataset. RefCOCO+ was specifically designed to focus on appearance-based descriptions rather than relational or contextual ones. Therefore, methods that isolate the target region (like Masked Crop) can perform well on this particular dataset. However, MaskInversion shows superior performance to Masked Crop across all other datasets and larger model architectures, particularly in scenarios requiring contextual understanding (PhraseCut and RefCOCO). We will clarify this dataset-specific characteristic in the paper.
>
> ## Minor correction:
> Thank you for catching the typo ("maks"). We will correct this in the final version.

---

> ### Author Response · Authors · 2024-11-26
> **Feedback on our response**
>
> We hope our detailed responses have addressed all your concerns. If you have no further questions, we would greatly appreciate if you could consider increasing the score.

---

> > ### Comment · Reviewer_rbTp · 2024-11-30
> >
> > Thank you for the detailed response and clarification. Most of my questions are now resolved, and I’ve updated my score accordingly.

---

> > > ### Author Response · Authors · 2024-12-02
> > >
> > > Thank you for your positive feedback and thoughtful evaluation of our work. We greatly appreciate your supportive comments indicating that our responses have addressed your concerns.
> > >
> > > We noticed that the review score hasn't been updated yet, and we would be grateful if you could kindly consider adjusting it to reflect your positive assessment.

---

### Official Review · Reviewer_zd2G · 2024-11-04

**Soundness:** 3
**Presentation:** 3
**Contribution:** 3
**Rating:** 6
**Confidence:** 4

**Summary:**

The paper introduces MaskInversion, a method designed to generate localized embeddings for specific image regions using pre-trained vision-language foundation models like CLIP. This approach leverages the feature representations of these models to create context-aware embeddings for a query image region specified by a mask at test time.

**Strengths:**

1. This paper is overall well-written.
2. The paper provides a comprehensive set of experiments and results, including quantitative metrics and qualitative visualizations, which helps in understanding the method's effectiveness and behavior.
3. MaskInversion operates in a zero-shot setting, which means it can handle tasks without requiring additional training data for specific tasks, leveraging the knowledge embedded in pre-trained models.

**Weaknesses:**

1. This paper may be a bit short on innovation, as it actually uses the explainability map obtained from LeGrad to improve the feature extraction of the pre-trained models. Besides, some of the methods section is devoted to reviewing LeGrad, reinforcing the perception that this article is not innovative enough.
2. The regulaization loss seems very important to avoid trivial solutions. However, I find no ablation study on the hyper-paramter $\alpha$, which modulates the influence of the regularization loss.
3. The performance of MaskInversion is heavily dependent on the quality of the input masks. In practical applications, obtaining high-quality masks might be challenging, which could limit the method's real-world applicability.
4. The paper could benefit from a deeper analysis of scenarios where MaskInversion might fail or underperform, and how such cases could be addressed.

**Questions:**

If the pre-trained model itself does not have strong local feature capture capability, then post-training can give limited improvement. I'm curious if this idea of mask-guided feature capture can be applied to the training phase to improve the fine-grained perception of pre-trained VL models.

---

> ### Author Response · Authors · 2024-11-22
>
> We thank the reviewer for their thorough and constructive feedback. We particularly appreciate the recognition of the paper's clear writing, comprehensive experimental evaluation, and the advantages of our zero-shot approach that leverages pre-trained models effectively.
>
> # Response to Weaknesses:
> ## 1. Innovation and LeGrad Description
> We understand the concern about the extensive description of LeGrad potentially overshadowing our contributions. We included this detailed explanation to ensure the paper is self-contained and accessible. We emphasize that our key innovation lies not in the use of LeGrad itself, but in the novel approach of using explainability maps to guide the learning of localized embeddings without any fine-tuning (Sec. 3.2). As a second contribution, we also propose a gradient decomposition strategy for efficient computation (Sec. 3.3). Following the reviewer’s comment, we offer to move the detailed LeGrad description to the supplementary material upon request, while maintaining only the essential components in the main text.
>
> ## 2. Regularization Loss Analysis
> Thanks for raising this topic. We want to point out that we see the regularization as a special feature that can be useful if there is a reason to include more global information in the LET (localized embedding token), which so far seems to be only relevant in cases of referential expressions. To further analyze the influence of $\alpha$, we conducted a respective ablation study on the RefCOCO dataset (please see **Table 6** in Annex of the current version of the paper). We further provide a qualitative example of the effect of alpha on the explainability map and the generated caption in the supplementary of the current version in **Figure 5**.
> The added figure illustrates this effect through generated captions for different $\alpha$ values. When $\alpha=0$, the model generates descriptions focused strictly on the masked region  (_e.g._, "woman in a boat"), while increasing $\alpha$ progressively incorporates more contextual information(_e.g._, "produce" or "vegetables")
>
> The results of both the quantitative and qualitative analysis (please see the updated annex written in blue) show that $\alpha$ acts as a "slider" controlling the balance between local and global information. When $\alpha$ is very small (≈0), the model focuses strictly on the masked region, potentially missing contextual cues. As  $\alpha$ increases, the model incorporates more spatial context, with optimal performance around  $\alpha=5.0$. At very high values ( $\alpha$>7.5), the performance slightly decreases as the representation becomes too similar to the global [CLS] token.
>
> ## 3. Performance depends on the quality of input masks
> We acknowledge this important practical consideration and have thoroughly investigated it in Sec. 4.5 (Table 4) of our paper. Our analysis shows that:
> Even using just bounding boxes instead of masks only results in a modest performance drop (44.7% to 42.9% on MSCOCO for the Class Retrieval Task).
> Automatically segmenting bounding boxes with SAM and using the resulting masks as input to our method achieves comparable performance to inputting ground-truth masks (45.0% vs 44.7%). Hence, in combination with SAM, our method works very well given cheap bounding-box input from the users, a practical application scenario.
> Our method is more sensitive to under-specification (erosion) of masks (42.7%).
>
> ## 4. Failure Cases
> Thank you for highlighting the need for deeper analysis of failure cases. We assume the main scenario where MaskInversion may underperform might be in case of resolution limitations. Namely, as the method's effectiveness is bounded by the grid size of the underlying foundation model, if masks, e.g., of very small objects, are below the regular grid sampling size, we will not be able to exactly recover the localized embedding. We have conducted an additional experiment where we disentangle the performance of MaskInversion depending on the size of the mask. The reported numbers are the retrieval accuracy on the COCO dataset obtained using ViT-B/16.
>
> | Size Category | Percentage |
> |--------------|------------|
> | Small (<10%) | 42.3 |
> | Medium (10-30%) | 62.5 |
> | Large (>30%) | 63.2 |
> | Overall | 44.7 |
>
> The table shows that MaskInversion performance is mainly bounded by the performance on small objects. We will add a respective discussion in the main paper.
> ## Response to Questions
> Your suggestion about incorporating mask-guided feature capture during training is interesting. We believe this could be implemented as a form of self-distillation during the pre-training phase to enhance the model's fine-grained perception capabilities. This represents an exciting direction for future work that could potentially improve the foundation model's local feature capture abilities directly. We appreciate this suggestion and will explore it in future research.

---

> > ### Author Response · Authors · 2024-12-02
> >
> > As the discussion period is nearing its end, we wanted to respectfully follow up on our responses to your valuable feedback. We have carefully addressed all raised concerns and added substantial improvements to our paper.
> >
> > We would greatly appreciate if you could review our responses and consider updating your assessment if you find our revisions satisfactory.
> > Thank you for your time and expertise

---

> ### Author Response · Authors · 2024-11-26
> **Feedback on our response**
>
> We thank you for your constructive feedback and have addressed all points in detail. If you have no further questions, we kindly ask you to consider increasing the score.

---

### Official Review · Reviewer_gqG2 · 2024-11-05

**Soundness:** 2
**Presentation:** 2
**Contribution:** 2
**Rating:** 5
**Confidence:** 5

**Summary:**

This paper introduces MaskInversion, a method that leverages pre-trained vision-language models (such as CLIP) to generate context-aware embeddings for specific image regions by optimizing explainability maps. It aims to improve localized image representation tasks, such as referring expression comprehension and captioning, while employing a gradient decomposition strategy to reduce computation.

The contributions of this paper include:
1) a new method that is able to learn localized embeddings for given queries;
2) an efficient gradient decomposition approach for multi-query masks;
3) improved performance on various downstream tasks.

**Strengths:**

1. The motivation is interesting. The problem of poor localization capabilities does exist in CLIP.
2. The proposed method is intuitive.
3. The performance is good. MaskInversion achieves superior results on a wide range of vision-language benchmarks.

**Weaknesses:**

1. Very important baselines are missing. I noticed that you have discussed the paper of MaskCLIP [1] but did not compare with it in the experiments. Actually, CLIP's localization issues can be addressed in a very simple way. You just need to reform the last layer's self-attention in the fasion of MaskCLIP (removing Q and K), SCLIP [2] (Q-to-Q and K-to-K attention), or CLIPSurgery [3] (V-to-V attention with dual paths). I believe by simply modifying CLIP with these methods (they are all training-free), the performance can be improved by a very large margin.

2. Given these baselines are missing, it's difficult to evaluate whether the new method is effective enough. As MaskInversion involves a much more complex process, I expect it to perform significantly better than those three baselines.

3. The other contribution of the paper, gradient decomposition, is not that significant. As shown in Table 5, It makes clear speed improvements only if we have >10 masks/image. What is the general case of the number of masks involved in your tasks?

4. Minor comments: there are some typos in the paper such as in Line 481, what does Table 4.5 refer to?

[1] Extract free dense labels from clip, in ECCV 2022

[2] SCLIP: rethinking self-attention for dense vision-language inference, in ECCV 2024

[3] A closer look at the explainability of contrastive language-image pre-training.

**Questions:**

See Weaknesses.

---- updates after rebuttal ----
I appreciate the authors' response and additional experiments for the metioned baselines. While MaskInversion outperforms the training-free approaches in most cases, some of my concerns are addressed. Howerver, the authors did not discuss the new results in the revised paper, which may cause misleading for readers. Overall, I still think this is a boarderline paper and have changed the score to 5. I still have concerns about the scalability of the method, as on OpenImagesV7, which is relatively more complex and has more masks in the images, MaskInversion performs worse than CLIPSurgery.

---

> ### Author Response · Authors · 2024-11-22
>
> We sincerely thank the reviewer for their thorough and constructive feedback. We particularly appreciate the suggestion to compare with MaskCLIP, SCLIP, and CLIPSurgery, which are indeed relevant baselines and strengthen the paper.
> ## 1. Missing Baselines:
> We thank the reviewer for bringing attention to these important baselines. Following your suggestion, we have conducted comprehensive experiments comparing MaskInversion with MaskCLIP [1], CLIPSurgery[2], and SCLIP[3] across all evaluation datasets. To compare the respective performance, we used the training-free pipelines to compute respective patch token representations and average pool all patch tokens inside the mask to get the localized embedding for that mask. Using the official implementation provided by the authors, we ran each method on our evaluation suite without any retraining, as these methods are designed to be training-free. We keep the same evaluation pipeline as in the paper and only change the localized embedding tokens used. The results are summarized in the table below. We report the top-1 Accuracy for all datasets extending Table 1 and Table 2 in the original paper:
>
>
>
> | Method | Backbone | VOC | Context | COCO | PhraseCut | RefCOCO | RefCOCO+ | OpenImagesV7 |
> |--------|----------|-----|---------|------|-----------|----------|-----------|--------------|
> | MaskCLIP | B/16 | 74.9 | 43.0 | 40.2 | 53.9 | 49.3 | 52.6 | 45.6 |
> | CLIPSurgery | B/16 | 70.8 | 53.5 | 41.7 | 52.5 | 48.9 | 52.0 | **49.5** |
> | SCLIP | B/16 | 64.3 | 43.0 | 33.4 | 37.2 | 40.7 | 42.4 | 45.5 |
> | **Ours** | B/16 | **85.4** | **58.1** | **44.7** | **57.2** | **56.1** | **58.3** | 46.3 |
> | MaskCLIP | L/14 | 55.1 | 33.2 | 29.3 | 47.6 | 43.2 | 47.2 | 32.5 |
> | CLIPSurgery | L/14 | 78.3 | 46.4 | 47.7 | 47.2 | 47.3 | 50.9 | 45.5 |
> | SCLIP | L/14 | 43.0 | 24.9 | 25.9 | 19.0 | 32.8 | 32.5 | 38.3 |
> | **Ours** | **L/14** | **91.0** | **59.0** | **56.0** | **60.2** | **56.1** | **60.2** | **48.7** |
> | MaskCLIP | H/14 | 61.8 | 37.8 | 30.9 | 45.9 | 34.6 | 39.6 | 36.9 |
> | CLIPSurgery | H/14 | 68.0 | 40.8 | 40.1 | 41.5 | 43.2 | 46.7 | 45.8 |
> | SCLIP | H/14 | 38.2 | 20.7 | 19.8 | 15.2 | 20.7 | 20.7 | 35.6 |
> | **Ours** | H/14 | **93.5** | **61.8** | **63.7** | **64.0** | **61.2** | **65.0** | **51.2** |
>
> Our results demonstrate that MaskInversion consistently outperforms these baselines across nearly all datasets and backbone sizes, with particularly strong improvements for larger backbones.
> It shows that only in one case (ViT-B/16) ClipSurgery is able to outperform our method on OpenImages V7. More importantly, it shows that MaskInversion consistently outperforms these baselines and profits from larger backbones by showing increased performance.
> In contrast, the performance of the evaluated training-free methods starts to degrade.
>
> While this shows the capabilities of MaskInversion, we want to emphasize that we consider MaskInversion and training-free methods as two different valuable lines of work that overlap in this task, but might be used in different contexts.
>
> [1] https://github.com/chongzhou96/MaskCLIP
> [2] https://github.com/wangf3014/SCLIP
> [3] https://github.com/xmed-lab/CLIP_Surgery
>
> ## 2. Complexity compared to training-free methods:
> We think that it is difficult to directly compare the trade-off between complexity and performance of MaskInversion compared to training-free methods as we think of them as two independent lines of work.
> _e.g._, while our method operates on the model outputs without requiring architectural modifications, ClipSurgery needs the modification of the forward-pass over several layers.
> If the reviewer has any suggestions on how to explore this topic further, please let us know.
>
>
> ## 3. Gradient Decomposition Significance:
> We appreciate the concern about the gradient decomposition's practical utility.
> While the speed improvements become significant only with >10 masks per image, which might not be common in e.g., human interactions, we believe this contribution is rather valuable for an automatic processing scenario. Indeed, recent methods such as SAM and open-world objectness detectors typically generate 100-250 masks per image. Being able to  efficiently process large amounts of masks might allow converting such image regions into meaningful embeddings. We consider this to be an interesting direction for future work.
>
>
> ## 4. Minor Comments:
> Thank you for catching the typo regarding Table 4.5 which is actually Table 5. This will be corrected in the final version.
>
> We believe the results and clarifications we provide in this rebuttal address the main concerns while demonstrating the significant advantages of our approach. We will update the paper to include these comparisons and clarify the practical significance of the gradient decomposition contribution.

---

> > ### Author Response · Authors · 2024-12-02
> >
> > As the discussion period is nearing its end, we wanted to respectfully follow up on our responses to your valuable feedback. We have carefully addressed all raised concerns and added substantial improvements to our paper.
> >
> > We would greatly appreciate if you could review our responses and consider updating your assessment if you find our revisions satisfactory.
> > Thank you for your time and expertise

---

> ### Author Response · Authors · 2024-11-26
> **Feedback on our response**
>
> We hope that our comprehensive response, particularly the extensive experimental comparisons with MaskCLIP, SCLIP, and CLIPSurgery, has adequately addressed your concerns about missing baselines and demonstrated the significant advantages of our approach. Given these new results showing consistent improvements across datasets and backbone sizes, we kindly ask if you would consider revising your assessment of our paper, or if you have any additional questions we could address.

---

### Meta-Review · Area_Chair_BC6P · 2024-12-21

**Metareview:**

This paper addresses the poor localization problem of contrastive image-text pretraining (CLIP) models, which is a critical issue when using CLIP models in practice. The paper is well-written and motivated, and experiments show improved performance on the target localization tasks.

However, as Reviewer gqG2 pointed out, the paper lacks comparison and discussion with training-free methods (MaskCLIP, SCLIP, and CLIPSurgery, as the reviewer suggested) that share the same (or similar) objective (i.e., resolving poor localization ability of CLIP). In addition, the paper lacks a discussion about additional costs compared to these training-free approaches. While the authors showed additional comparisons with those baselines in their rebuttal, these results were not reflected in the revised version of the paper, which remains the same concerns from the initial review.

Initial reviews also highlighted concerns about complex design, lack of ablation studies, and required optimization steps compared to other baselines. Additionally, the AC agrees with Reviewer zd2G that the methodology heavily relies on LeGrad's explainability method and lacks technical novelty.

The AC considers this a borderline paper with both strengths and weaknesses. Given the reviews and rebuttal, and the highly competitive nature of ICLR submissions, the AC recommends rejection.

**Additional Comments On Reviewer Discussion:**

Reviewer gqG2 raised issues about missing baselines (MaskCLIP, SCLIP, CLIPSurgery). The authors answered with some experimental results, but they did not update the paper accordingly. The AC finds that the authors failed to address the reviewer's concerns clearly.

---

### Decision · Program_Chairs · 2025-01-22

Reject